# The Effect of a Horse-Riding Simulator with Virtual Reality on Gross Motor Function and Body Composition of Children with Cerebral Palsy: Preliminary Study

**DOI:** 10.3390/s22082903

**Published:** 2022-04-10

**Authors:** Yong Gi Jung, Hyun Jung Chang, Eun Sol Jo, Da Hye Kim

**Affiliations:** 1Department of Otorhinolaryngology-Head and Neck Surgery, Samsung Medical Center, School of Medicine, Sungkyunkwan University, Seoul 06351, Korea; ent.jyg@gmail.com; 2Department of Physical Medicine and Rehabilitation, Samsung Changwon Hospital, School of Medicine, Sungkyunkwan University, Changwon 51353, Korea; orange200000@naver.com (E.S.J.); dahae2005@naver.com (D.H.K.)

**Keywords:** cerebral palsy, virtual reality, horse-riding simulator, body composition, rehabilitation

## Abstract

This study aimed to evaluate the effect of a horse-riding simulator (HRS) with virtual reality (VR) on gross motor function, balance control, and body composition in children with spastic cerebral palsy (CP). Seventeen preschool and school-aged children with spastic CP were included; 10 children in the intervention group (HRS group) received 30 min of HRS with VR training twice a week for a total of 16 sessions in addition to conventional physiotherapy. Seven children in the control group were instructed to perform home-based aerobic exercises twice a week for 8 weeks in addition to conventional physiotherapy. Gross motor function measure (GMFM) and body composition were evaluated before the first session and after the last session. Before and after the 2-month intervention, Pediatric Balance Scale and Timed Up and Go test were evaluated for the HRS group. GMFM scores and body composition changed significantly in the HRS group (*p* < 0.05). However, no significant differences were observed in the control group. Changes in the GMFM total scores, GMFM dimension D scores, and skeletal muscle mass significantly differed between the HRS and control groups (*p* < 0.05). HRS with VR may be an effective adjunctive therapeutic approach for the rehabilitation of children with CP.

## 1. Introduction

Cerebral palsy (CP), one of the most common physical disabilities in childhood, is a disorder of movement and posture caused by non-progressive lesions in the developing brain [1]. Children with CP, to varying degrees, have muscle weakness, tone abnormality, and motor-control impairment, causing abnormal posture and poor balance control. Rehabilitative treatment was administered to improve the symptoms.

Hippotherapy is an equine-assisted therapy that applies specific movements of horses during rehabilitation [2]. According to a systematic review of interventions for children with CP, hippotherapy is an effective motor intervention for improving balance and symmetry; it positively affects spasticity, gross motor function, and hand function [3]. Therefore, hippotherapy should be administered to children with CP. However, this is often not always available due to distance, weather, and cost. A horse-riding simulator (HRS) is an intervention based on hippotherapy consisting of a robotic device with a dynamic saddle that imitates the movement of a riding horse by producing three-dimensional movements similar to the horse gait pattern [4]. In a systematic review of interventions for children with CP, intervention with HRS showed a positive effect on postural balance, gross motor function, and hand function in children with CP [3]. Therefore, HRS can be used as an alternative to hippotherapy; however, HRS is reportedly less effective [5]. Another reason other than using a robotic horse instead of a real horse is the difference in the level of sensory stimulation. If a technology that can promote various sensory stimulation is applied to HRS, it can be expected to achieve similar effects to hippotherapy.

Virtual reality (VR) is a computer-generated simulation that provides a feeling of being in a real environment with the help of three-dimensional images [6]. VR technology provides an opportunity for users to manipulate and experience objects in the virtual environment. The use of VR in physiotherapy and rehabilitation has significantly increased over the last decade. Two recent meta-analyses of randomized controlled trials showed that VR games have a positive effect on the improvement of gross motor skills and balance in children with CP [7,8]. In addition, a systematic review reported that the positive effects of VR with gaming on gross motor and hand function were augmented when combined with task-specific motor training [3]. VR enables difficult movement to be performed in a secure environment. If HRS is used in conjunction with VR, the children would feel as if they were riding a real horse, which enhances the therapeutic effect of HRS by providing multiple postural challenges for them.

Children with CP have body composition alterations with increased FM and low FFM across the spectrum of all functional capacity. Gross motor function and physical activity are closely related to altered body composition. Noble et al. reported that the increase in muscle mass relative to body mass growth is reduced in individuals with CP compared to that in their typically developing peers [9]. The trajectory of muscle growth in CP may be related to the severity of motor disability and may affect long-term mobility. For long-term mobility, children with CP should perform appropriate physical activity to improve body composition, particularly to increase muscle mass. Previous studies have reported that muscle mass and strength increased after HRS intervention in adults [10,11]. Therefore, it can be expected that HRS could improve body composition in children with CP. However, no studies have investigated the effects of HRS on body composition in children with CP.

The objective of this study was to evaluate the effect of HRS with VR on gross motor function, balance control, and body composition in preschool- and school-aged children with CP.

## 2. Materials and Methods

### 2.1. Equipment and Program Development

An HRS with VR was developed for the rehabilitation of children with CP. It consists of a horse-riding simulator (Shinhwa EQ-900, Seoul, Korea) with a safety harness and a head-mounted display with a pair of controllers (Odyssey, Samsung Electronics, Suwon, Korea) for an immersive VR experience (Figure 1). Upper extremity activity and trunk motion applied to VR simulation were precisely designed and tuned by a pediatric rehabilitation doctor (H.J.C.). In addition, playtime and difficulty in the VR scenario were developed in consultation with the rehabilitation team considering the condition of the children participating in the study Following the development of HRS with VR, 10 typically developing children (TDC) experienced riding on the HRS with VR before clinical trials for children with CP. No adverse events such as falls, VR sickness, or pain occurred.

### 2.2. Participants

This study was conducted after the protocol was approved by the Institutional Review Board of Samsung Changwon Hospital (SCMC 2020-04-001). The participants in the study and their parents or legal guardians signed informed consent forms after a detailed explanation about the study.

The inclusion criteria were as follows: (1) diagnosed as spastic CP; (2) aged between 5 and 18 years; (3) classified in level I-IV of the Gross Motor Function Classification System (GMFCS); and (4) able to sit astride on a saddle. The exclusion criteria were as follows: (1) having received an injection of botulinum toxin within 3 months; (2) having undergone orthopedic surgery or selective dorsal rhizotomy in the previous 1 year; (3) having undergone selective dorsal rhizotomy; (4) having undergone HRS training within 6 months; (5) having severe pain, joint dislocation, contracture, or spinal deformity; (6) having experienced uncontrolled epileptic seizure; (7) having moderate-to-severe intellectual disability; and (8) having poor visual or hearing acuity.

### 2.3. Procedure

Children in the intervention group (HRS group) underwent HRS sessions with VR training and conventional physiotherapy. Thirty-minute training sessions were conducted twice a week for a total of 16 sessions. Each session consisted of a 2-min warm-up, one cycle consisting of an 11-min training program, a 3-min rest period, one cycle consisting of an 11-min training program, and a 1-min cool-down. The VR training program was structured as follows. On starting the program, the riders first watched the short instruction video. Next, their arm reach was measured, and they could simultaneously adjust to sitting on the saddle with their hands lifted. During training, the riders hit the target by raising their arms and avoiding obstacles by tilting their trunks laterally on a moving saddle. Vital signs such as blood pressure, pulse rate, and body temperature were checked before and after each riding session.

Children on the waiting list served as controls. They were instructed to perform the home-based aerobic exercise twice a week for 8 weeks, in addition to conventional physiotherapy.

### 2.4. Outcome Measure

The primary outcome was measured as the change in the percentage of gross motor function measure (GMFM). The GMFM is used to evaluate changes that occur over time in the gross motor function of children with CP [12,13]. GMFM-88 consists of the original 88 items, each scored on a scale of 0–3, and is grouped into five dimensions: (A) lying and rolling (17 items); (B) sitting (20 items); (C) crawling and kneeling (14 items); (D) standing (13 items); and (E) walking, running, and jumping (24 items). The sum of the scores of each item is the score for each dimension, which was expressed as a percentage of the maximum score for each dimension. TDC can generally accomplish all items of the GMFM up to the age of 5 years [12]. The GMFM-88 was administered to all participants before the first session and after the last session by the same physical therapist blinded to the analysis. GMFM-66 scores were calculated from GMFM-88 using the Gross Motor Ability Estimator. The GMFM-66 consists of a subset of the GMFM-88 items identified to quantitatively represent the gross motor function in children with CP through Rasch analysis. The GMFM-66 was somewhat shorter and scaled compared to GMFM-88 [14]. Both GMFM-88 and GMFM-66 are evaluative outcome measures to detect changes in gross motor function with interventions in children with CP [13].

The secondary outcome measure was body composition using a commercially available bioelectrical impedance analysis (BIA) called InBody S10 (Biospace, Seoul, Korea) [15]. Inbody S10 is based on multifrequency BIA, which helps analyze body composition in five segments of the body at six different frequencies (1, 5, 50, 250, 500, and 1000 kHz). The BIA method is widely used in clinical practice because of its ready accessibility, low cost, and quick assessment. There was a strong correlation between the measurement of body composition using dual-energy X-ray absorptiometry (DXA) and BIA with InBody S10 [16]. For standardization of measurement conditions, subjects were asked to fast for 2–3 h, void, and restrict physical activity before measurement [17]. In terms of posture, the arm was abducted by 30°, and the legs were separated by 45°. In addition, the examiner cleaned the skin with alcohol and attached electrodes of 4 cm^2^ or more to an area of the skin without lesions or significant edema.

Before and after the 2-month intervention, the Pediatric Balance Scale (PBS) and Timed Up and Go (TUG) test were evaluated for the intervention group, if possible.

PBS was used to assess balance among children in the HRS group [18]. It comprises 14 items that assess the functional activities necessary to safely and independently perform daily activities at home, school, or in the community, such as sit-to-stand, stand-to-sit, transfers, sitting balance, standing balance, 1 leg standing, turning, reaching the floor, reaching forward, and stepping on and off an elevated surface. Each item is scored on a scale of 0–4. The TUG test is a practical and straightforward objective measurement of functional mobility [19]. It is time a person takes to rise from a chair, walk 3 m, turn around 180°, walk back to the chair, and sit down.

### 2.5. Statistical Analysis

The statistical analyses were performed using SPSS 21.0 (SPSS Inc., Chicago, IL, USA). The chi-square test was used to test whether distributions of categorical variables among the baseline characteristics between the two groups differ from each other. The Mann–Whitney test was used to compare continuous variables among the baseline characteristics, initial states of participants and changes in outcome measures between the two groups. Within-group changes before and after intervention were assessed using the Wilcoxon signed-rank test. The statistical significance was considered at *p* < 0.05.

## 3. Results

Seventeen preschool- and school-aged children with CP and GMFCS levels I–IV were included, with 10 children in the HRS group (males, 7; females, 3) and seven children in the control group (males, 4; females, 3) (Table 1). The mean age of children was 112.1 months in the HRS group and 109.0 months in the control group. No significant differences in mean age, gender distribution, GMFCS level, GMFM score, and body composition were observed between the two groups at baseline.

The GMFM-66, GMFM-88 total, and GMFM dimensions D and E scores increased significantly after intervention in the HRS group (*p* < 0.05) (Table 2). However, no significant differences were observed between the scores of pre-intervention and post-intervention in the control group. Changes in the GMFM-66, GMFM-88 total scores, and GMFM dimension D scores significantly differed between the HRS and control groups (*p* < 0.05). 

Height, fat-free mass, and skeletal muscle mass (SMM) significantly increased in the HRS group after intervention (Table 3). There was no significant change in the body mass index. Body weight and fat mass decreased, but without statistical significance. Some children had increased fat mass probably because diet control was not mandatory. Encouragingly, SMM increased in all children in the HRS group without exception. However, no significant changes were observed in the control group. The HRS group showed increased SMM compared to the control group (*p* < 0.05).

After the intervention, PBS and TUG scores showed significant improvement (Figure 2). No adverse events, such as falls, VR sickness, or pain were observed.

## 4. Discussion

This is the first study to evaluate the improvement of body composition and motor performance in children with CP by applying fully immersive VR technology to HRS and has high clinical significance. This study showed that HRS with VR had beneficial effects by improving gross motor function, balance control, mobility, and body composition among preschool- and school-aged children with spastic CP without serious adverse events.

All the children in the HRS group showed an increase in GMFM-66 score, but not all the children in the control group. The control group showed no significant changes in the GMFM. It has been reported that children with CP, on average, reach approximately 90% of their motor function by 5 years or younger, depending on their GMFCS level [20]. The curves appear to plateau approximately 7 years before functional decline [21]. The age of the children included in this study was 5 years or older, and there was no improvement in gross motor function with previous treatment alone. In contrast, the intervention group showed functional improvement. Damiano suggested that physical activity and exercise are essential for maintaining function and performance in persons with CP [22]. Therefore, it can be inferred that HRS with VR provides appropriate physical activity for children with CP to improve function.

Since the beginning of 2020, the coronavirus disease (COVID-19) has continued worldwide. Children with CP have impairments in pulmonary function and are therefore at a high risk of respiratory complications from COVID-19 [23]. Therefore, they need to practice physical distancing and minimize visits to public places. Most children with CP did not undergo routine check-ups during the COVID-19 pandemic, and some dropped out of physical therapy sessions [24]. Research reported that more than half of the children with CP had increased tonus, decreased range of motion, decreased physical activities, and decreased rehabilitation services during the pandemic period [25]. In contrast, physical activity levels, home programs, and environmental support positively affected body function. Unlike hippotherapy or other rehabilitation treatments, HRS with VR does not require contact with many people. Moreover, HRS with VR can be integrated into a home program as part of telerehabilitation. Therefore, HRS with VR can provide adequate physical activity while social distancing.

Few studies have evaluated the effect of HRS in children with CP using the GMFM as an evaluation tool. Herrero et al. reported that HRS did not improve GMFM in children with CP [26]. In contrast, our study showed that the GMFM-88 total score increased significantly after the intervention. Compared to Herrero’s study, the main difference is that our study incorporated VR, which provides multiple directional challenges and various sensory stimulations. VR provides an opportunity for active learning and intrigues, encourages participants, and ensures motivation [6]. Motivation has been suggested to be an important factor in pediatric motor rehabilitation [27]. Motivation and attention are vital modulators of neuroplasticity. A successful task-specific practice is rewarding and enjoyable for children, producing spontaneous, regular practice. Therefore, motivated children have better rehabilitation outcomes than unmotivated children. Thus, HRS combined with VR was more effective than HRS alone. Furthermore, it is expected that HRS with VR can produce similar effects to hippotherapy. However, further research comparing the effects of HRS with those of VR and hippotherapy is needed.

Several previous studies have investigated the effect of HRS on balance, especially on sitting balance [5,26,28]. These studies reported that the HRS produced significant improvements in the postural control of children with CP in the sitting posture. Our study showed that the mean value of GMFM dimension B (sitting) increased without statistical significance. It should be noted that the functional level of the subjects in those studies was poorer than that of the subjects in our study. In one study, most subjects belonged to GMFCS V. Many of the children included in our study belonged to GMFCS I. Therefore, it can be considered that there was no improvement in the sitting dimension due to the ceiling effect and heterogeneity of subject function in this study. However, the GMFM dimensions D and E, representative of standing activity, increased significantly after the intervention. In addition, the PBS score showed significant improvement. Previous studies have reported that HRS improves balance and motor performance in children with CP. The postulated mechanism by which HRS improves balance and motor function is as follows [5]. Reactive trunk control can be improved as the protective mechanism to maintain the posture without falling is activated when children with CP try to sit with a balance on a moving saddle. And body perception in space is also facilitated. While shifting body weight in response to a moving surface, multiple sensory inputs and reactive motor outputs are stimulated. Immediate motor response to various stimuli and efforts to maintain balance and trunk-upright posture can improve postural stability, equilibrium reaction, and correction of upright alignment.

In this study, children who underwent HRS with VR showed significant improvements in body composition, such as increased height, fat-free mass, and SMM. In preschool-aged children with CP, the poorer the function, the higher the percentage of fat mass and the lower the fat-free mass. [29]. Children with spastic CP have lower height percentile and physical activity level, and more adipose tissue infiltration of skeletal muscle compared to TDC [30]. Children with CP spend more time sedentary than TDC but significantly less time participating in moderate-to-vigorous activities [31]. The time spent on moderate-to-vigorous activity was inversely related to the fat mass. Adequate physical activity is vital for improving body composition and health status. However, children with more severe motor impairments have more difficulty participating in physical activity. In several studies on the effects of HRS, children with GMFCS grade V were included. HRS is an activity in which children with severe motor impairments can participate. A study on the energy consumption while riding an HRS with VR is necessary to determine the activity level.

This study had some limitations. The sample size was small. The functional levels of the participants were heterogeneous. Therefore, a randomized controlled trial with a large sample size and a comparison of the effects according to the GMFCS level would be needed. DXA is a reproducible and reliable technique for measuring body composition. However, DXA is a costly device and carries the risk of exposure to radiation [16]. Since this is a preliminary study to determine whether HRS with VR improves body composition, cost-effective BIA was used for body composition analysis. BIA measurements were performed according to the guidelines to reduce errors as much as possible [17]. Further studies using DXA are required. We suggest that HRS combined with VR is more effective than is HRS alone. However, it is difficult to judge the effects of VR accurately using this study alone; further research comparing the effects of HRS alone and HRS with VR is required. 

## 5. Conclusions

HRS with VR may be an effective therapeutic approach for the rehabilitation of children with CP. It can improve motor function, mobility, and balance control and can be expected to help improve body composition. Based on the clinical effects and technological advances demonstrated in this study, it can be expanded to various rehabilitation treatment fields and serve as a foundation to increase patient compliance and treatment efficacy.

## Figures and Tables

**Figure 1 sensors-22-02903-f001:**
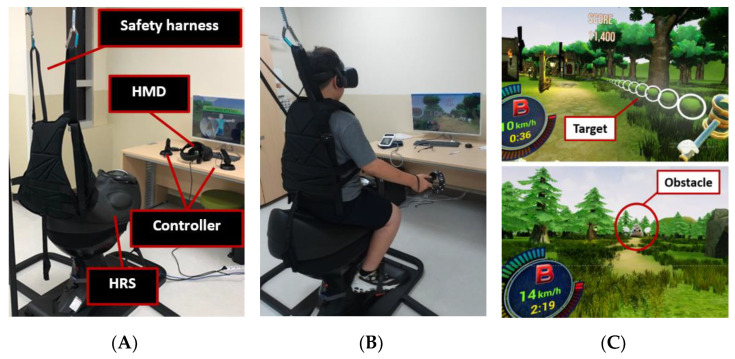
System of a horse-riding simulator (HRS) with virtual reality (VR) (**A**) shows an HRS with a safety harness and head-mounted display (HMD) with controllers. (**B**,**C**) During training, the target is hit by raising the arms, and obstacles are avoided by tilting the trunk laterally on a moving saddle.

**Figure 2 sensors-22-02903-f002:**
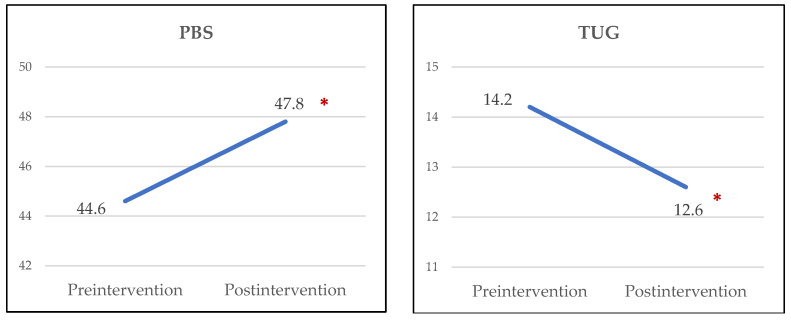
Changes in Pediatric Balance Scale (PBS) and Timed Up and Go (TUG) test. * Statistically significant difference between pre- and postintervention (*p* < 0.05).

**Table 1 sensors-22-02903-t001:** Characteristics of the participants.

Characteristics		HRS Group (*n* = 10)	Control Group (*n* = 7)	*p*-Value
Age (Months)		112.1 ± 25.3	109.0 ± 29.0	0.82
Gender	Male	7	4	0.64
	Female	3	3	
GMFCS level	I	6	4	0.83
	II	1	1	
	III	1		
	IV	2	2	
Topography	Unilateral	6	4	0.91
	Bilateral	4	3	

Age values are presented as mean ± standard deviation. GMFCS, gross motor function classification system.

**Table 2 sensors-22-02903-t002:** Changes in gross motor function measures of the intervention (HRS) and control groups.

	HRS Group (*n* = 10)			Control Group (*n* = 7)			*p*-Value for Difference between Group
GMFM	Preintervention	Postintervention	*p*-Value	Preintervention	Postintervention	*p*-Value	
A	95.1 ± 9.7	96.9 ± 5.4	0.32	97.5 ± 4.6	97.5 ± 4.6	0.99	0.74
B	90.2 ± 20.0	91.2 ± 17.6	0.18	89.8 ± 20.2	89.8 ± 20.2	0.99	0.54
C	84.5 ± 29.2	86.7 ± 26.0	0.32	84.0 ± 29.7	83.0 ± 29.6	0.18	0.23
D	72.6 ± 35.8	74.4 ± 35.6	0.03	68.1 ± 38.1	67.8 ± 37.6	0.71	0.06
E	68.6 ± 39.6	69.3 ± 39.7	0.03	66.8 ± 43.2	66.5 ± 42.6	0.68	0.19
GMFM-88 total	82.2 ± 26.1	83.5 ± 24.3	<0.01	81.2 ± 26.7	80.9 ± 26.5	0.25	<0.01
GMFM-66	73.4 ± 20.1	75.3 ± 21.7	<0.01	71.4 ± 20.8	70.6 ± 20.1	0.14	<0.01

Values are presented as mean ± standard deviation. GMFM, gross motor function measure; A, lying and rolling; B, sitting; C, crawling and kneeling; D, standing; E, walking, running, and jumping.

**Table 3 sensors-22-02903-t003:** Changes in body composition of the intervention (HRS) and control groups.

	HRS Group (*n* = 10)			Control Group (*n* = 7)			*p*-Value for Difference between Group
	Preintervention	Postintervention	*p*-Value	Preintervention	Postintervention	*p*-Value	
Height	1.28 ± 0.19	1.30 ± 0.20	0.01	1.27 ± 0.13	1.28 ± 0.13	0.99	0.42
Weight	34.9 ± 18.1	35.4 ± 17.4	0.07	31.3 ± 9.4	31.4 ± 9.4	0.99	0.03
BMI	19.9 ± 4.8	19.5 ± 4.2	0.48	18.9 ± 3.2	18.7 ± 3.2	0.99	0.74
FM	9.5 ± 7.1	8.1 ± 6.4	0.10	7.6 ± 4.7	7.5 ± 4.8	>0.99	0.19
FFM	25.4 ± 11.5	27.3 ± 11.5	<0.01	23.7 ± 5.6	23.9 ± 5.6	0.74	0.07
SMM	13.0 ± 6.8	14.2 ± 6.8	<0.01	12.0 ± 3.3	12.1 ± 3.3	0.80	0.04
BF	24.5 ± 11.1	19.8 ± 9.8	0.04	22.3 ± 9.9	22.1 ± 10.0	>0.99	0.23

Values are presented as mean ± standard deviation. BMI, body mass index. FM, fat mass. FFM, fat-free mass. SMM, skeletal muscle mass. BF, percent body fat.

## Data Availability

The data presented in this study are available on request from the corresponding author.

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
