# Peer review of "The Effect of a Horse-Riding Simulator with Virtual Reality on Gross Motor Function and Body Composition of Children with Cerebral Palsy: Preliminary Study"

_sensors, 2022, doi:10.3390/s22082903_

Round 1

Reviewer 1 Report

The paper reports on the study of telerehabilitation practices based on a Horse-Riding  Simulator combined with VR.

The results are really very interesting and the proposed methodology seems to work very well.

Reviewer 2 Report

Overall, the manuscript needs restructuring to increase its readability. It may need to provide stronger justification to make it more relevant and interesting to the readers of the journal.

Section 2.1 Program development – the protocol of the study (starting in line 81) session should be presented in a separate section as ‘Procedure.’

Section 2.2 Participants – the basic information of the sample is not presented here (e.g., number of participants in each condition, the distribution of gender, age).

Section 2.3 Interventions – the authors can consider renaming this section as ‘Procedure’.

The abstract is not written concisely. Some information like motivation, background, and implications of the results are lacking.

The readability of the introduction section is relatively low. It is not easy to appreciate the motivation, significance, and contribution of the reported study. Surprisingly, the additional background information is provided in the discussion section (starting in line 218) instead of the introduction section.

Language issue(s):

The first line of the abstract does not seem to be a complete sentence.

Reviewer 3 Report

The novelty of the proposed research paper is highlighted by the developed horse-riding simulator that integrates an Odyssey head-mount display with a pair of controllers. The originality of the research is related to the intended VR application applied to analyze the effect on gross motor function of children with cerebral palsy.

The research presented in the proposed paper is significant for the development of medical rehabilitation applications as it presents a case study of a custom designed Horse-Riding Simulation that makes use of a safety harness and a HMD with controllers to evaluate the gross motor functions, balance control and body composition in children with spastic cerebral palsy. With the recent development of VR equipment and sensors, many researchers have started to design VR application intended for medical rehabilitation.

The paper is well documented with a good amount of recent related works references. The paper has an extended introduction that provides background research related to cerebral palsy as it is one of the most common physical disabilities in childhood. The authors provide additional information regarding hippotherapy, as it is one of the most effective motor interventions to improve balance and symmetry. As presented by the authors applying hippotherapy to children with CP is more complicated as it involves aspects related to distance, availability it is also influenced by weather and the costs are significant. The authors decided to create a cost-efficient robotic device that integrates a dynamic saddle that will imitate the movement associated with hippotherapy. The objective of the paper is clearly presented at the end of the introduction as a research study aimed to evaluate the effect of Horse-Riding Simulation with VR integration in preschool and school-aged children with cerebral palsy.

The materials and methods section presents the proposed simulator as well as the associated VR application that requires participants to hit various targets by raising their arms and avoid obstacles by tilting the trunk laterally on the robotic saddle. The following subsection provides details regarding the seventeen preschool and school-aged children with spastic CP included in the proposed study. The children in the intervention group underwent HRS sessions with VR training and conventional physiotherapy. Details regarding the outcome measure within the research are presented within the following subsection as well as the statistical analysis associated with the research findings. The results section is organized using three tables and a figure that illustrate the changes in pediatric balance scale and timed-up and go test, as presented by the authors there is a statistically significant difference between pre and post intervention of the proposed rehabilitation robotic devices that integrates a VR equipment. The discussions are based on the authors finding and are compared to other related work previous studies that investigate the effect of hippotherapy on balance and sitting balance. The current limitations of the studies are well defined and considering the small sample size it is hard to highlight the effects of the VR system using this study alone. The authors intend to study the effects of HRS alone and HRS with VR in the future.

The proposed research work presents accurate results regarding the evaluation of the proposed Horse-Riding Simulator that integrates a VR headset and controller on gross motor function, balance control, and body composition of children with cerebral palsy.

The research paper has a high interest for readers and scientists that are developing medical rehabilitation devices that integrate robotic structures paired with commercially available VR system to design and implement cost-effective rehabilitation systems.

The authors have designed and evaluated a cost-effective rehabilitation system aimed to evaluate the effects of the proposed horse-riding simulator on gross motor function and body composition of children with cerebral palsy.

Round 2

Reviewer 2 Report

The manuscript has been improved. Thank you.